# Understanding Why Generalized Reweighting Does Not Improve Over ERM

## Abstract

Empirical risk minimization (ERM) is known to be non-robust in practice to distributional shift where the training and the test distributions are different. A suite of approaches, such as importance weighting, and variants of distributionally robust optimization (DRO), have been proposed to solve this problem. But a line of recent work has empirically shown that these approaches do not significantly improve over ERM in real applications with distribution shift. The goal of this work is to obtain a comprehensive theoretical understanding of this intriguing phenomenon. We first posit the class of Generalized Reweighting (GRW) algorithms, as a broad category of approaches that iteratively update model parameters based on iterative reweighting of the training samples. We show that when overparameterized models are trained under GRW, the resulting models are close to that obtained by ERM. We also show that adding small regularization which does not greatly affect the empirical training accuracy does not help. Together, our results show that a broad category of what we term GRW approaches are not able to achieve distributionally robust generalization. Our work thus has the following sobering takeaway: to make progress towards distributionally robust generalization, we either have to develop non-GRW approaches, or perhaps devise novel classification/regression loss functions that are adapted to the class of GRW approaches.

## 1 Introduction

It has now been well established that empirical risk minimization (ERM) can empirically achieve high test performance on a variety of tasks, particularly with modern overparameterized models where the number of parameters is much larger than the number of training samples. This strong performance of ERM however has been shown to degrade under *distributional shift*, where the training and test distributions are different [HS15, BGO16, Tat17]. There are two broad categories of distribution shift: *domain generalization* where the test distribution contains new *environments* not in the training distribution like in domain adaptation, and *subpopulation shift* where the two distributions have the same set of subpopulations but their mixture weights differ like in algorithmic fairness applications.

People have proposed various approaches to learn models that are robust to distributional shift. The most classical approach is importance weighting (IW) [Shi00], which reweights training samples; in the context of subpopulation shift these weights are typically set so that each subpopulation/group has the same overall weight in the training objective. The approach most widely used today is Distributional Robust Optimization (DRO) [DN18, HSNL18], in which we assume that the test distribution belongs to a certain set of distributions that are close to the training distribution (called the *uncertainty set*), and train the model on the worst distribution in that set. Many variants of DRO have been proposed and are used in practice [HNSS18, SKHL20, XDKR20, ZDKR21, ZDS$^+$21].

While these approaches have been developed for the express purpose of improving ERM for distribution shift, a line of recent work has empirically shown the negative result that when used to train overparameterized models, these methods do not improve over ERM. For IW, [BL19] observed that its effect under stochastic gradient descent (SGD) diminishes over training epochs, and finally does not improve over ERM. For variants of DRO, [SKHL20] found that these methods overfit very easily, i.e. their test performances will drop to the same low level as ERM after sufficiently many epochs if no regularization is applied. [GLP21, KSM$^+$21] compared these methods with ERM on a number of real-world applications, and found that in most cases none of these methods improves over ERM.

This line of empirical results has also been bolstered by some recent theoretical results. [SRKL20] constructed a synthetic dataset where a linear model trained with IW is provably not robust to subpopulation shift. [XYR21] further proved that under gradient descent (GD) with a sufficiently small learning rate, a linear classifier trained with either IW or ERM converges to the same max-margin classifier, and thus upon convergence, are no different. These previous theoretical results are limited to linear models and specific approaches such as IW where sample weights are fixed during training. They are not applicable to more complex models, and more general approaches where the sample weights could iteratively change, including most DRO variants.

Towards placing the empirical results on a stronger theoretical footing, we define the class of *generalized reweighting* (GRW), which dynamically assigns weights to the training samples, and iteratively minimizes the weighted average of the sample losses. By allowing the weights to vary with iterations, we cover not just static importance weighting, but also DRO approaches outlined earlier; though of course, the GRW class is much broader than just these instances.

In this work, we prove the comprehensive result that in both regression and classification, and for both overparameterized linear models and wide neural networks, the models learnt via any GRW approach and ERM are similar, in the sense that their implicit biases are (almost) equivalent. We note that extending the analysis from linear models to wide neural networks is non-trivial since it requires the result that wide neural networks can be approximated by their linearized counterparts to hold *uniformly throughout the iterative process* of GRW algorithms. Our results extend the analysis in [LXS$^+$19], but as we show, the proof in the original paper had some flaws, and due to which we have to fix the proof by changing the network initialization (Eqn. (9), see Appendix E).

Overall, the important takeaway is that *distributionally robust generalization* cannot be directly achieved by the broad class of GRW algorithms (which includes popular approaches such as importance weighting and most DRO variants). Progress towards this important goal thus requires either going beyond GRW algorithms, or devising novel loss functions that are adapted to GRW approaches. In Section 6 we will discuss some promising future directions as well as the limitations of this work.

## 2  Preliminaries

Let the input space be $\mathcal{X} \subseteq \mathbb{R}^d$ and the output space be $\mathcal{Y} \subseteq \mathbb{R}$.[1] We assume that $\mathcal{X}$ is a subset of the unit $L_2$ ball of $\mathbb{R}^d$, so that any $\boldsymbol{x} \in \mathcal{X}$ satisfies $\|\boldsymbol{x}\|_2 \leq 1$. We have a training set $\{\boldsymbol{z}_i = (\boldsymbol{x}_i, y_i)\}_{i=1}^n$ *i.i.d.* sampled from an underlying distribution $P$ over $\mathcal{X} \times \mathcal{Y}$. Denote $\boldsymbol{X} = (\boldsymbol{x}_1, \cdots, \boldsymbol{x}_n) \in \mathbb{R}^{d \times n}$, and $\boldsymbol{Y} = (y_1, \cdots, y_n) \in \mathbb{R}^n$. For any function $g : \mathcal{X} \mapsto \mathbb{R}^m$, we overload notation and use $g(\boldsymbol{X}) = (g(\boldsymbol{x}_1), \cdots, g(\boldsymbol{x}_n)) \in \mathbb{R}^{m \times n}$ (except when $m = 1$, $g(\boldsymbol{X})$ is defined as a column vector). Let the loss function be $\ell : \mathcal{Y} \times \mathcal{Y} \to [0, 1]$. ERM trains a model by minimizing its *expected risk* $\mathcal{R}(f; P) = \mathbb{E}_{\boldsymbol{z} \sim P}[\ell(f(\boldsymbol{x}), y)]$ via minimizing the *empirical risk* $\hat{\mathcal{R}}(f) = \frac{1}{n} \sum_{i=1}^n \ell(f(\boldsymbol{x}_i), y_i)$.

In distributional shift, the model is evaluated not on the training distribution $P$, but a different test distribution $P_{\text{test}}$, so that we care about the expected risk $\mathcal{R}(f; P_{\text{test}})$. A large family of methods designed for such distributional shift is *distributionally robust optimization* (DRO), which minimizes the expected risk over the worst-case distribution $Q \ll P$[2] in a ball w.r.t. divergence $D$ around the training distribution $P$. Specifically, DRO minimizes the *expected DRO risk* defined as:

$$\mathcal{R}_{D,\rho}(f; P) = \sup_{Q \ll P} \{\mathbb{E}_Q[\ell(f(\boldsymbol{x}), y)] : D(Q \parallel P) \leq \rho\} \tag{1}$$

for $\rho > 0$. Examples include CVaR, $\chi^2$-DRO [HSNL18], and DORO [ZDKR21], among others.

---

[1]Our results can be easily extended to the multi-class scenario (see Appendix B).

[2]For distributions $P$ and $Q$, $Q$ is *absolute continuous* to $P$, or $Q \ll P$, means that for any event $A$, $P(A) = 0$ implies $Q(A) = 0$.

A common category of distribution shift is known as subpopulation shift. Let the data domain contain $K$ *groups* $\mathcal{D}_1, \cdots, \mathcal{D}_K$. The training distribution $P$ is the distribution over all groups, and the test distribution $P_{\text{test}}$ is the distribution over one of the groups. Let $P_k(\boldsymbol{z}) = P(\boldsymbol{z} \mid \boldsymbol{z} \in \mathcal{D}_k)$ be the conditional distribution over group $k$, then $P_{\text{test}}$ can be any one of $P_1, \cdots, P_k$. The goal is to train a model $f$ that performs well over every group. There are two common ways to achieve this goal: one is minimizing the *balanced empirical risk* which is an unweighted average of the empirical risk over each group, and the other is minimizing the *worst-group risk* defined as

$$\mathcal{R}_{\max}(f; P) = \max_{k=1,\cdots,K} \mathcal{R}(f; P_k) = \max_{k=1,\cdots,K} \mathbb{E}_{\boldsymbol{z} \sim P}[\ell(f(\boldsymbol{x}), y) | z \in \mathcal{D}_k] \tag{2}$$

# 3   Generalized Reweighting (GRW)

Various methods have been proposed towards learning models that are robust to distributional shift. In contrast to analyzing each of these individually, we instead consider a large class of what we call Generalized Reweighting (GRW) algorithms that includes the ones mentioned earlier, but potentially many others more. Loosely, GRW algorithms iteratively assign each sample a weight during training (that could vary with the iteration) and iteratively minimize the weighted average risk. Specifically, at iteration $t$, GRW assigns a weight $q_i^{(t)}$ to sample $\boldsymbol{z}_i$, and minimizes the weighted empirical risk:

$$\hat{\mathcal{R}}_{\boldsymbol{q}^{(t)}}(f) = \sum_{i=1}^{n} q_i^{(t)} \ell(f(\boldsymbol{x}_i), y_i) \tag{3}$$

where $\boldsymbol{q}^{(t)} = (q_1^{(t)}, \cdots, q_n^{(t)})$ and $q_1^{(t)} + \cdots + q_n^{(t)} = 1$.

*Static GRW* assigns to each $\boldsymbol{z}_i = (\boldsymbol{x}_i, y_i)$ a fixed weight $q_i$ that does not change during training, i.e. $q_i^{(t)} \equiv q_i$. A classical method is *importance weighting* [Shi00], where if $\boldsymbol{z}_i \in \mathcal{D}_k$ and the size of $\mathcal{D}_k$ is $n_k$, then $q_i = (K n_k)^{-1}$. Under importance weighting, (3) becomes the balanced empirical risk in which each group has the same weight. Note that ERM is also a special case of static GRW.

On the other hand, in *dynamic GRW*, $\boldsymbol{q}^{(t)}$ changes with $t$. For instance, any approach that iteratively upweights samples with high losses in order to help the model learn "hard" samples, such as DRO, is an instance of GRW. When estimating the population DRO risk $\mathcal{R}_{D,\rho}(f; P)$ in Eqn. (1), if $P$ is set to the empirical distribution over the training samples, then $Q \ll P$ implies that $Q$ is also a distribution over the training samples. Thus, DRO methods belong to the broad class of GRW algorithms. There are two common ways to implement DRO. One uses Danskin's theorem and chooses $Q$ as the maximizer of $\mathbb{E}_Q[\ell(f(\boldsymbol{x}), y)]$ in each epoch. The other one formulates DRO as a bi-level optimization problem, where the lower level updates the model to minimize the expected risk over $Q$, and the upper level updates $Q$ to maximize it. Both can be seen as instances of GRW. As one popular instance of the latter, *Group DRO* was proposed by [SKHL20] to minimize (2). Denote the empirical risk over group $k$ by $\hat{\mathcal{R}}_k(f)$, and the model at time $t$ by $f^{(t)}$. Group DRO iteratively sets $q_i^{(t)} = g_k^{(t)}/n_k$ for all $\boldsymbol{z}_i \in \mathcal{D}_k$ where $g_k^{(t)}$ is the group weight that is updated as

$$g_k^{(t)} \propto g_k^{(t-1)} \exp\left(\nu \hat{\mathcal{R}}_k(f^{(t-1)})\right) \ (\forall k = 1, \cdots, K) \tag{4}$$

for some $\nu > 0$, and then normalized so that $q_1^{(t)} + \cdots + q_n^{(t)} = 1$. [SKHL20] then showed (in their Proposition 2) that for convex settings, the Group DRO risk of iterates converges to the global minimum with the rate $O(t^{-1/2})$ if $\nu$ is sufficiently small.

# 4   Theoretical Results for Regression

In this section, we will study GRW for regression tasks that use the squared loss

$$\ell(\hat{y}, y) = \frac{1}{2}(\hat{y} - y)^2. \tag{5}$$

We will prove that for both linear models and sufficiently wide fully-connected neural networks, the implicit bias of GRW is equivalent to ERM, so that starting from the same initial point, GRW and ERM will converge to the same point when trained for an infinitely long time, which explains why GRW does not improve over ERM without regularization and early stopping. We will further show that while regularization can affect this implicit bias, it must be large enough to *significantly lower the training performance*, or the final model will still be similar to the unregularized ERM model.

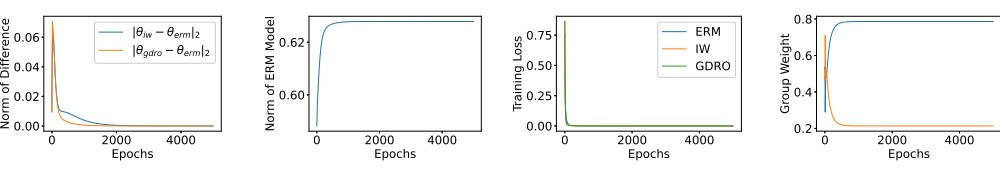

(a) Weight Difference     (b) Norm of ERM Model     (c) Training Loss     (d) Group Weights

Figure 1: Experimental results of ERM, importance weighting (IW) and Group DRO (GDRO) with the squared loss on six MNIST images with a linear model. All norms are $L_2$ norms.

## 4.1 Linear Models

We first demonstrate our result on simple linear models to provide our readers with a key intuition; later, we will apply this same intuition to neural networks. This key intuition draws from results of [GLSS18]. Let the linear model be denoted by $f(\boldsymbol{x}) = \langle \theta, \boldsymbol{x} \rangle$, where $\theta \in \mathbb{R}^d$. We consider the overparameterized setting where $d > n$. The weight update rule of GRW under GD is the following:

$$\theta^{(t+1)} = \theta^{(t)} - \eta \sum_{i=1}^{n} q_i^{(t)} \nabla_\theta \ell(f^{(t)}(\boldsymbol{x}_i), y_i) \tag{6}$$

where $\eta > 0$ is the learning rate. For a linear model with the squared loss, the update rule is

$$\theta^{(t+1)} = \theta^{(t)} - \eta \sum_{i=1}^{n} q_i^{(t)} \boldsymbol{x}_i (f^{(t)}(\boldsymbol{x}_i) - y_i) \tag{7}$$

For this training scheme, we can prove that if the training error converges to zero, then the model converges to an interpolator $\theta^*$ (s.t. $\forall i, \langle \theta^*, \boldsymbol{x}_i \rangle = y_i$) independent of $q_i^{(t)}$ (proofs in Appendix D):

**Theorem 1.** *If $\boldsymbol{x}_1, \cdots, \boldsymbol{x}_n$ are linearly independent, then under the squared loss, for any GRW such that the empirical training risk $\hat{\mathcal{R}}(f^{(t)}) \to 0$ as $t \to \infty$, it holds that $\theta^{(t)}$ converges to an interpolator $\theta^*$ that only depends on $\theta^{(0)}$ and $\boldsymbol{x}_1, \cdots, \boldsymbol{x}_n$, but does not depend on $q_i^{(t)}$.*

The proof is based on the following key intuition regarding the update rule (7): $\theta^{(t+1)} - \theta^{(t)}$ is a linear combination of $\boldsymbol{x}_1, \cdots, \boldsymbol{x}_n$ for all $t$, so $\theta^{(t)} - \theta^{(0)}$ always lies in the linear subspace $\text{span}\{\boldsymbol{x}_1, \cdots, \boldsymbol{x}_n\}$, which is an $n$-dimensional linear subspace if $\boldsymbol{x}_1, \cdots, \boldsymbol{x}_n$ are linearly independent. By Cramer's rule, there is exactly one $\tilde{\theta}$ in this subspace such that we get interpolation of all the data $\langle \tilde{\theta} + \theta^{(0)}, \boldsymbol{x}_i \rangle = y_i$ for all $i \in \{1, \ldots, n\}$. In other words, the parameter $\theta^* = \tilde{\theta} + \theta^{(0)}$ in this subspace that interpolates all the data is unique. Thus the proof would follow if we were to show that $\theta^{(t)} - \theta^{(0)}$, which lies in the subspace, also converges to interpolating the data.

We have essentially proved the following sobering result: *the implicit bias of any GRW that achieves zero training error is equivalent to ERM, so GRW does not improve over ERM*. While the various distributional shift methods discussed in the introduction have been shown to satisfy the precondition of convergence to zero training error with overparameterized models and linearly independent inputs [SKHL20], we provide the following theorem that shows this for the broad class of GRW methods. Specifically, we show this result for any GRW satisfying the following assumption with a sufficiently small learning rate:

**Assumption 1.** There are constants $q_1, \cdots, q_n$ s.t. $\forall i, q_i^{(t)} \to q_i$ as $t \to \infty$. And $\min_i q_i = q^* > 0$.

**Theorem 2.** *If $\boldsymbol{x}_1, \cdots, \boldsymbol{x}_n$ are linearly independent, then there exists $\eta_0 > 0$ such that for any GRW satisfying Assumption 1 with the squared loss, and any $\eta \le \eta_0$, the empirical training risk $\hat{\mathcal{R}}(f^{(t)}) \to 0$ as $t \to \infty$.*

Finally, we use a simple experiment to demonstrate the correctness of this result. The experiment is conducted on a training set of six MNIST images, five of which are digit 0 and one is digit 1. We use a 784-dimensional linear model and run ERM, importance weighting and group DRO. The results are presented in Figure 1, and they show that the training loss of each method converges to 0, and the gap between the model weights of importance weighting, Group DRO and ERM converges to 0, meaning that all three model weights converge to the same point, whose $L_2$ norm is about 0.63. Figure 1d also shows that the group weights in Group DRO empirically satisfy Assumption 1.

## 4.2 Wide Neural Networks (Wide NNs)

Now we study *sufficiently wide fully-connected neural networks*. We extend the analysis in [LXS+19] in the neural tangent kernel (NTK) regime [JGH18]. In particular we study the following network:

$$\boldsymbol{h}^{l+1} = \frac{W^l}{\sqrt{d_l}}\boldsymbol{x}^l + \beta\boldsymbol{b}^l \qquad \text{and} \qquad \boldsymbol{x}^{l+1} = \sigma(\boldsymbol{h}^{l+1}) \qquad (l = 0, \cdots, L) \tag{8}$$

where $\sigma$ is a non-linear activation function, $W^l \in \mathbb{R}^{d_{l+1} \times d_l}$ and $W^L \in \mathbb{R}^{1 \times d_L}$. Here $d_0 = d$. The parameter vector $\theta$ consists of $W^0, \cdots, W^L$ and $b^0, \cdots, b^L$ ($\theta$ is the concatenation of all flattened weights and biases). The final output is $f(\boldsymbol{x}) = \boldsymbol{h}^{L+1}$. And let the neural network be initialized as

$$\begin{cases} W_{i,j}^{l(0)} \sim \mathcal{N}(0,1) \\ \boldsymbol{b}_j^{l(0)} \sim \mathcal{N}(0,1) \end{cases} (l = 0, \cdots, L-1) \qquad \text{and} \qquad \begin{cases} W_{i,j}^{L(0)} = 0 \\ \boldsymbol{b}_j^{L(0)} \sim \mathcal{N}(0,1) \end{cases} \tag{9}$$

We also need the following assumption on the wide neural network:

**Assumption 2.** $\sigma$ is differentiable everywhere. Both $\sigma$ and its first-order derivative $\dot{\sigma}$ are Lipschitz.[3]

**Difference from [JGH18].** Our initialization (9) differs from the original one in [JGH18] in the last (output) layer, where we use the zero initialization $W_{i,j}^{L(0)} = 0$ instead of the Gaussian initialization $W_{i,j}^{L(0)} \sim \mathcal{N}(0,1)$. This modification permits us to accurately approximate the neural network with its linearized counterpart (11), as we notice that the proofs in [LXS+19] (particularly the proofs of their Theorem 2.1 and their Lemma 1 in Appendix G) are flawed. In Appendix E we will explain what goes wrong in their proofs and how we manage to fix the proofs with our modification.

Denote the neural network at time $t$ by $f^{(t)}(\boldsymbol{x}) = f(\boldsymbol{x}; \theta^{(t)})$ which is parameterized by $\theta^{(t)} \in \mathbb{R}^p$ where $p$ is the number of parameters. We use the shorthand $\nabla_\theta f^{(0)}(\boldsymbol{x}) := \nabla_\theta f(\boldsymbol{x}; \theta)\big|_{\theta=\theta_0}$. The *neural tangent kernel* (NTK) of this model is $\Theta^{(0)}(\boldsymbol{x}, \boldsymbol{x}') = \nabla_\theta f^{(0)}(\boldsymbol{x})^\top \nabla_\theta f^{(0)}(\boldsymbol{x}')$, and the *Gram matrix* is $\Theta^{(0)} = \Theta^{(0)}(\boldsymbol{X}, \boldsymbol{X}) \in \mathbb{R}^{n \times n}$. For this wide NN, we still have the following NTK theorem:

**Lemma 3.** *If $\sigma$ is Lipschitz and $d_l \to \infty$ for $l = 1, \cdots, L$ sequentially, then $\Theta^{(0)}(\boldsymbol{x}, \boldsymbol{x}')$ converges in probability to a non-degenerate[4] deterministic limiting kernel $\Theta(\boldsymbol{x}, \boldsymbol{x}')$.*

The *kernel Gram matrix* $\Theta = \Theta(\boldsymbol{X}, \boldsymbol{X}) \in \mathbb{R}^{n \times n}$ is a positive semi-definite symmetric matrix. Denote its largest and smallest eigenvalues by $\lambda^{\max}$ and $\lambda^{\min}$. Note that $\Theta$ is non-degenerate, so we can assume that $\lambda^{\min} > 0$ (which is almost surely true when $d_L \gg n$). Then we have:

**Theorem 4.** *Let $f^{(t)}$ be a wide fully-connected neural network that satisfies Assumption 2 and is trained by any GRW satisfying Assumption 1 with the squared loss. Let $f_{\mathrm{ERM}}^{(t)}$ be the same model trained by ERM from the same initial point. If $d_1 = \cdots = d_L = \tilde{d}$, $\nabla_\theta f^{(0)}(\boldsymbol{x}_1), \cdots, \nabla_\theta f^{(0)}(\boldsymbol{x}_n)$ are linearly independent, and $\lambda^{\min} > 0$, then there exists a constant $\eta_1 > 0$ such that: if $\eta \leq \eta_1$[5], then for any $\delta > 0$, there exists $\tilde{D} > 0$ such that as long as $\tilde{d} \geq \tilde{D}$, with probability at least $(1 - \delta)$ over random initialization we have: for any test point $\boldsymbol{x} \in \mathbb{R}^d$ such that $\|\boldsymbol{x}\|_2 \leq 1$, as $\tilde{d} \to \infty$,*

$$\limsup_{t \to \infty} \left| f^{(t)}(\boldsymbol{x}) - f_{\mathrm{ERM}}^{(t)}(\boldsymbol{x}) \right| = O(\tilde{d}^{-1/4}) \to 0 \tag{10}$$

Note that for simplicity, in the theorem we only consider the case where $d_1 = \cdots = d_L = \tilde{d} \to \infty$, but in fact the result can be very easily extended to the case where $d_l/d_1 \to \alpha_l$ for $l = 2, \cdots, L$ for some constants $\alpha_2, \cdots, \alpha_L$, and $d_1 \to \infty$. Here we provide a proof sketch for this theorem. The key is to consider the *linearized neural network* of $f^{(t)}(\boldsymbol{x})$:

$$f_{\mathrm{lin}}^{(t)}(\boldsymbol{x}) = f^{(0)}(\boldsymbol{x}) + \langle \theta^{(t)} - \theta^{(0)}, \nabla_\theta f^{(0)}(\boldsymbol{x}) \rangle \tag{11}$$

which is a linear model with features $\nabla_\theta f^{(0)}(\boldsymbol{x})$. Thus if $\nabla_\theta f^{(0)}(\boldsymbol{x}_1), \cdots, \nabla_\theta f^{(0)}(\boldsymbol{x}_n)$ are linearly independent, then the linearized NN converges to the unique interpolator. Then we show that the

---

[3] $f$ is *Lipschitz* if there exists a constant $L > 0$ such that for any $\boldsymbol{x}_1, \boldsymbol{x}_2, |f(\boldsymbol{x}_1) - f(\boldsymbol{x}_2)| \leq L\|\boldsymbol{x}_1 - \boldsymbol{x}_2\|_2$.
[4] *Non-degenerate* means that $\Theta(\boldsymbol{x}, \boldsymbol{x}')$ depends on $\boldsymbol{x}$ and $\boldsymbol{x}'$ and is not a constant.
[5] For ease of understanding, later we will write this condition as "with a sufficiently small learning rate".

wide neural network can be approximated by its linearized counterpart *uniformly throughout training*, which is considerably more subtle in our case due to the GRW dynamics. Here we prove that the gap is bounded by $O(\tilde{d}^{-1/4})$, but in fact we can prove that it is bounded by $O(\tilde{d}^{-1/2+\epsilon})$ for any $\epsilon > 0$:

**Lemma 5** (Approximation Theorem). *For a wide fully-connected neural network $f^{(t)}$ satisfying Assumption 2 and is trained by any GRW satisfying Assumption 1 with the squared loss, let $f_{\text{lin}}^{(t)}$ be its linearized neural network trained by the same GRW (i.e. $q_i^{(t)}$ are the same for both networks for any $i$ and $t$). Under the conditions of Theorem 4, with a sufficiently small learning rate, for any $\delta > 0$, there exist constants $\tilde{D} > 0$ and $C > 0$ such that as long as $\tilde{d} \geq \tilde{D}$, with probability at least $(1-\delta)$ over random initialization we have: for any test point $\boldsymbol{x} \in \mathbb{R}^d$ such that $\|\boldsymbol{x}\|_2 \leq 1$,*

$$\sup_{t \geq 0} \left| f_{\text{lin}}^{(t)}(\boldsymbol{x}) - f^{(t)}(\boldsymbol{x}) \right| \leq C\tilde{d}^{-1/4} \tag{12}$$

Theorem 4 shows that at *any test point $\boldsymbol{x}$* within the unit ball, the gap between the outputs of wide NNs trained by GRW and ERM from the same initial point is arbitrarily close to 0. So we have shown that for regression, with both linear and wide NNs, GRW does not improve over ERM.

## 4.3 Wide Neural Networks, with $L_2$ Regularization

Previous work such as [SKHL20] proposed to improve DRO algorithms by adding $L_2$ penalty to the objective function. In this section, we thus study adding $L_2$ regularization to GRW algorithms:

$$\hat{\mathcal{R}}_{\boldsymbol{q}^{(t)}}^{\mu}(f) = \sum_{i=1}^{n} q_i^{(t)} \ell(f(\boldsymbol{x}_i), y_i) + \frac{\mu}{2} \left\| \theta - \theta^{(0)} \right\|_2^2 \tag{13}$$

From the outset, it is easy to see that under $L_2$ regularization, GRW methods have different implicit biases than ERM. For example, when $f$ is a linear model, $\ell$ is convex and smooth, then $\hat{\mathcal{R}}_{\boldsymbol{q}^{(t)}}^{\mu}(f)$ with static GRW is a convex smooth objective function, so under GD with a sufficiently small learning rate, the model will converge to the global minimizer (see Appendix D.1). Moreover, the global optimum $\theta^*$ satisfies $\nabla_\theta \hat{\mathcal{R}}_{\boldsymbol{q}^{(t)}}^{\mu}(f(\boldsymbol{x}; \theta^*)) = 0$, solving which yields $\theta^* = \theta^{(0)} + (\boldsymbol{X}\boldsymbol{Q}\boldsymbol{X}^\top + \mu\boldsymbol{I})^{-1}\boldsymbol{X}\boldsymbol{Q}(\boldsymbol{Y} - f^{(0)}(\boldsymbol{X}))$, which depends on $\boldsymbol{Q} = \text{diag}(q_1, \cdots, q_n)$, so adding $L_2$ regularization at least seems to yield different results from ERM (whether it improves over ERM might depend on $q_1, \cdots, q_n$).

However, the following result shows that this regularization must be large enough to *significantly lower the training performance*, or the resulting model would still be close to the unregularized ERM model. We still denote the largest and smallest eigenvalues of the kernel Gram matrix $\Theta$ by $\lambda^{\max}$ and $\lambda^{\min}$. We use the subscript "reg" to refer to a regularized model (trained by minimizing (13)).

**Theorem 6.** *Suppose there exists $M_0 > 0$ s.t. $\left\| \nabla_\theta f^{(0)}(\boldsymbol{x}) \right\|_2 \leq M_0$ for all $\|\boldsymbol{x}\|_2 \leq 1$. If $\lambda^{\min} > 0$ and $\mu > 0$, then for a wide NN satisfying Assumption 2, and any GRW minimizing the squared loss with a sufficiently small learning rate $\eta$, if $d_1 = d_2 = \cdots = d_L = \tilde{d}$, $\nabla_\theta f^{(0)}(\boldsymbol{x}_1), \cdots, \nabla_\theta f^{(0)}(\boldsymbol{x}_n)$ are linearly independent, and the empirical training risk of $f_{\text{reg}}^{(t)}$ satisfies*

$$\limsup_{t \to \infty} \hat{\mathcal{R}}(f_{\text{reg}}^{(t)}) < \epsilon \tag{14}$$

*for some $\epsilon > 0$, then with a sufficiently small learning rate, as $\tilde{d} \to \infty$, with probability close to 1 over random initialization, for any $\boldsymbol{x}$ such that $\|\boldsymbol{x}\|_2 \leq 1$ we have*

$$\limsup_{t \to \infty} \left| f_{\text{reg}}^{(t)}(\boldsymbol{x}) - f_{\text{ERM}}^{(t)}(\boldsymbol{x}) \right| = O(\tilde{d}^{-1/4} + \sqrt{\epsilon}) \to O(\sqrt{\epsilon}) \tag{15}$$

*where $f_{\text{reg}}^{(t)}$ is trained by regularized GRW and $f_{\text{ERM}}^{(t)}$ by unregularized ERM from same initial points.*

The proof again starts from analyzing linearized neural networks, and showing that regularization does not help there (Appendix D.4.2). Then, we need to prove a new approximation theorem for $L_2$ regularized GRW connecting wide NNs to their linearized counterparts uniformly through the GRW training process (Appendix D.4.1). Note that with regularization, we no longer need Assumption 1 to prove the new approximation theorem, because previously Assumption 1 is used to prove the convergence of GRW, but with regularization GRW naturally converges.

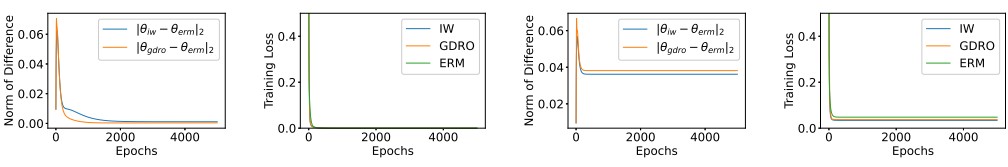

(a) Weight Difference     (b) Training Loss     (c) Weight Difference     (d) Training Loss

Figure 2: Experimental results of ERM, importance weighting (IW) and Group DRO (GDRO) with $L_2$ regularization with the squared loss. Left two: $\mu = 0.1$; Right two: $\mu = 10$.

Theorem 6 shows that if the training error can go below $\epsilon$, then the gap between the outputs of the two models *on any test point* $\boldsymbol{x}$ within the unit ball will be at most $O(\sqrt{\epsilon})$. Thus, if $\epsilon$ is very small, regularized GRW yields a very similar model to unregularized ERM, and thus makes improvement.

To empirically demonstrate this result, we run the same experiment as in Section 4.1 but with $L_2$ regularization. The results are presented in Figure 2. We can see that when the regularization is small, the training losses still converge to 0, and the three model weights still converge to the same point. On the contrary, with a large regularization, the training loss does not converge to 0, and the three model weights no longer converge to the same point. This shows that the regularization must be large enough to lower the training performance in order to make a significant difference to the implicit bias.

## 5   Theoretical Results for Classification

Now we consider classification where $\mathcal{Y} = \{+1, -1\}$. The big difference is that *classification losses don't have finite minimizers*. A classification loss converging to zero means that the model weight "explodes" to infinity instead of converging to a finite point. We focus on the canonical logistic loss:

$$\ell(\hat{y}, y) = \log(1 + \exp(-\hat{y}y)) \tag{16}$$

### 5.1   Linear Models

We first consider training the linear model $f(\boldsymbol{x}) = \langle \theta, \boldsymbol{x} \rangle$ with GRW under gradient descent with the logistic loss. As noted earlier, in this setting, [BL19] made the empirical observation that importance weighting does not improve over ERM. Then, [XYR21] proved that for importance weighting algorithms, as $t \to \infty$, $\|\theta^{(t)}\|_2 \to \infty$ and $\theta^{(t)}/\|\theta^{(t)}\|_2$ converges to a unit vector that does not depend on the sample weights, so it does not improve over ERM. To extend this theoretical result to the broad class of GRW algorithms, we will prove two results. First, in Theorem 7 we will show that under the logistic loss, any GRW algorithm satisfying the following weaker assumption:

**Assumption 3.** For all $i$, $\liminf_{t \to \infty} q_i^{(t)} > 0$,

if the training error converges to 0, and the direction of the model weight converges to a fixed unit vector, then this unit vector must be the *max-margin classifier* defined as

$$\hat{\theta}_{\mathrm{MM}} = \arg\max_{\theta : \|\theta\|_2 = 1} \left\{ \min_{i = 1, \cdots, n} y_i \cdot \langle \theta, \boldsymbol{x}_i \rangle \right\} \tag{17}$$

Second, Theorem 8 shows that for any GRW satisfying Assumption 1, the training error converges to 0 and the direction of the model weight converges, so it does not improve over ERM.

**Theorem 7.** *If* $\boldsymbol{x}_1, \cdots, \boldsymbol{x}_n$ *are linearly independent, then for the logistic loss, we have: for any GRW satisfying Assumption 3, if as* $t \to \infty$ *the empirical training risk* $\hat{\mathcal{R}}(f^{(t)})$ *converges to 0 and* $\theta^{(t)}/\|\theta^{(t)}\|_2 \to \boldsymbol{u}$ *for some unit vector* $\boldsymbol{u}$, *then* $\boldsymbol{u} = \hat{\theta}_{MM}$.

This result is an extension of [SHN+18]. Note that $\hat{\theta}_{\mathrm{MM}}$ does not depend on $q_i^{(t)}$, so this result shows that the sample weights have no effect on the implicit bias. Thus, for any GRW method that only satisfies the weak Assumption 3, as long as the training error converges to 0 and the model weight direction converges, GRW does not improve over ERM. We next show that any GRW satisfying Assumption 1 does have its model weight direction converge, and its training error converge to 0.

**Theorem 8.** *For any loss* $\ell$ *that is convex, $L$-smooth in* $\hat{y}$ *and strictly monotonically decreasing to zero as* $y\hat{y} \to +\infty$, *and GRW satisfying Assumption 1, denote* $F(\theta) = \sum_{i=1}^n q_i \ell(\langle \theta, \boldsymbol{x}_i \rangle, y_i)$. *If* $\boldsymbol{x}_1, \cdots, \boldsymbol{x}_n$ *are linearly independent, then with a sufficiently small learning rate* $\eta$, *we have:*

*(i)* $F(\theta^{(t)}) \to 0$ *as* $t \to \infty$.                    *(ii)* $\left\|\theta^{(t)}\right\|_2 \to \infty$ *as* $t \to \infty$.

*(iii) Let* $\theta_R = \arg\min_\theta\{F(\theta) : \|\theta\|_2 \le R\}$. $\theta_R$ *is unique for any* $R$ *such that* $\min_{\|\theta\|_2 \le R} F(\theta) <$
$\min_i q_i \ell(0, y_i)$. *And if* $\lim_{R \to \infty} \frac{\theta_R}{R}$ *exists, then* $\lim_{t \to \infty} \frac{\theta^{(t)}}{\left\|\theta^{(t)}\right\|_2}$ *also exists and they are equal.*

This result is an extension of Theorem 1 of [JDST20]. For the logistic loss, it is easy to show that
it satisfies the conditions of the above theorem and $\lim_{R \to \infty} \frac{\theta_R}{R} = \hat{\theta}_{\mathrm{MM}}$. Thus, Theorems 8 and 7
together imply that all GRW satisfying Assumption 1 (including ERM) have the same implicit bias
(see Appendix D.5.3). We also have empirical verification for these results (see Appendix C).

**Remark.**  It is impossible to extend these results to wide NNs like Theorem 4 because for a neural
network, if $\|\theta^{(t)}\|_2$ goes to infinity, then $\|\nabla_\theta f\|_2$ will also go to infinity. However, for a linear model,
the gradient is a constant. Consequently, the gap between the neural networks and its linearized
counterpart will "explode" under gradient descent, so there can be no approximation theorem like
Lemma 5 that can connect wide NNs to their linearized counterparts. Thus, we consider regularized
GRW, for which $\theta^{(t)}$ converges to a finite point and there is an approximation theorem.

## 5.2  Wide Neural Networks, with $L_2$ Regularization

Consider minimizing the regularized weighted empirical risk (13) with $\ell$ being the logistic loss. As in
the regression case, with $L_2$ regularization, GRW methods have different implicit biases than ERM
for the same reasons as in Section 4.3. And similarly, we can show that in order for GRW methods to
be sufficiently different from ERM, the regularization needs to be large enough to significantly lower
the training performance. Specifically, in the following theorem we show that if the regularization
is too small to lower the training performance, then a wide neural network trained with regularized
GRW and the logistic loss will still be very close to the *max-margin linearized neural network*:

$$f_{\mathrm{MM}}(\boldsymbol{x}) = \langle \hat{\theta}_{\mathrm{MM}}, \nabla_\theta f^{(0)}(\boldsymbol{x}) \rangle \quad \text{where} \quad \hat{\theta}_{\mathrm{MM}} = \arg\max_{\|\theta\|_2 = 1} \left\{ \min_{i=1,\cdots,n} y_i \cdot \langle \theta, \nabla_\theta f^{(0)}(\boldsymbol{x}_i) \rangle \right\} \quad (18)$$

Note that $f_{\mathrm{MM}}$ does not depend on $q_i^{(t)}$. Moreover, using the result in the previous section we can
show that a linearized neural network trained with unregularized ERM will converge to $f_{\mathrm{MM}}$:

**Theorem 9.** *Suppose there exists* $M_0 > 0$ *such that* $\left\|\nabla_\theta f^{(0)}(\boldsymbol{x})\right\|_2 \le M_0$ *for all test point* $\boldsymbol{x}$. *For a
wide NN satisfying Assumption 2, and for any GRW satisfying Assumption 1 with the logistic loss,
if* $d_1 = d_2 = \cdots = d_L = \tilde{d}$ *and* $\nabla_\theta f^{(0)}(\boldsymbol{x}_1), \cdots, \nabla_\theta f^{(0)}(\boldsymbol{x}_n)$ *are linearly independent and the
learning rate is sufficiently small, then for any* $\delta > 0$ *there exists a constant* $C > 0$ *such that: with
probability at least* $(1 - \delta)$ *over random initialization, as* $\tilde{d} \to \infty$ *we have: for any* $\epsilon \in (0, \frac{1}{4})$, *if
the empirical training error satisfies* $\limsup_{t \to \infty} \hat{\mathcal{R}}(f_{\mathrm{reg}}^{(t)}) < \epsilon$, *then for any test point* $\boldsymbol{x}$ *such that*
$|f_{MM}(\boldsymbol{x})| > C \cdot (-\log 2\epsilon)^{-1/2}$, $f_{\mathrm{reg}}^{(t)}(\boldsymbol{x})$ *has the same sign as* $f_{MM}(\boldsymbol{x})$ *when* $t$ *is sufficiently large.*

This result says that at any test point $\boldsymbol{x}$ on which the max-margin linear classifier classifies with a
margin of $\Omega((-\log 2\epsilon)^{-1/2})$, the neural network has the same prediction. And as $\epsilon$ decreases, the
confidence threshold also becomes lower. Similar to Theorem 6, this theorem provides the scaling of
the gap between the regularized GRW model and the unregularized ERM model *w.r.t.* $\epsilon$.

This result justifies the empirical observation in [SKHL20] that with large regularization, some GRW
algorithms can maintain a high worst-group test performance, with the cost of suffering a significant
drop in training accuracy. On the other hand, if the regularization is small and the model can achieve
nearly perfect training accuracy, then its worst-group test performance will still significantly drop.

## 6  Discussion

### 6.1  Distributionally Robust Generalization and Future Directions

A large body of prior work focused on distributionally robust optimization, but we show that these
methods have (almost) equivalent implicit biases as ERM. In other words, *distributionally robust
optimization* (DRO) does not necessarily have better *distributionally robust generalization* (DRG).

315 Therefore, we argue that it is necessary to design principled ways to improve DRG, which is what
316 people really want in the first place. Here we discuss three promising approaches to improving DRG.

317 The first approach is data augmentation and pretraining on large datasets. Our theoretical findings
318 suggest that the implicit bias of GRW is determined by the training samples and the initial point, but
319 not the sample weights. Thus, to improve DRG, we can either obtain more training samples, or start
320 from a better initial point, as demonstrated in two recent papers [WGS$^+$22, SKL$^+$22].

321 The second approach (for classification) is to go beyond the class of (iterative) sample reweighting
322 based GRW algorithms, for instance via *logit adjustment* [MJR$^+$21], which makes a classifier *have*
323 *larger margins on smaller groups* to improve its generalization on smaller groups. An early approach
324 by [CWG$^+$19] proposed to add an $O(n_k^{-1/4})$ additive adjustment term to the logits output by the
325 classifier. Following this spirit, [MJR$^+$21] proposed the LA-loss which also adds an additive adjust-
326 ment term to the logits. [YCZC20] proposed the CDT-loss which adds a multiplicative adjustment
327 term to the logits by dividing the logits of different classes with different temperatures. [KPOT21]
328 proposed the VS-loss which includes both additive and multiplicative adjustment terms, and they
329 showed that only the multiplicative adjustment term affects the implicit bias, while the additive term
330 only affects optimization, a fact that can be easily derived from our Theorem 8. Finally, [LZT$^+$21]
331 proposed AutoBalance which optimizes the adjustment terms with a bi-level optimization framework.

332 The third approach is to stay within the class of GRW algorithms, but to change the classifica-
333 tion/regression loss function to be suited to GRW. A recent paper [WCHH22] showed that for linear
334 classifiers, one can make the implicit bias of GRW dependent on the sample weights by replacing the
335 exponentially-tailed logistic loss with the following *polynomially-tailed loss*:

$$\ell_{\alpha,\beta}(\hat{y}, y) = \begin{cases} \ell_{\text{left}}(\hat{y}y) & \text{, if } \hat{y}y < \beta \\ \dfrac{1}{[\hat{y}y - (\beta-1)]^\alpha} & \text{, if } \hat{y}y \geq \beta \end{cases} \tag{19}$$

336 And this result can be extended to GRW satisfying Assumption 1 using our Theorem 8. The reason
337 why loss (19) works is that it changes $\lim_{R \to \infty} \frac{\theta_R}{R}$, and the new limit depends on the sample weights.

## 6.2 Limitations

339 Like most theory papers, our work makes some strong assumptions. The two main assumptions are:

340   (i) The model is a linear model or a sufficiently wide fully-connected neural network.

341   (ii) The model is trained for sufficiently long time, *i.e.* without early stopping.

342 Regarding (i), [COB19] argued that NTK neural networks fall in the "lazy training" regime and
343 results might not be transferable to general neural networks. However, this class of neural networks
344 has been widely studied in recent years and has provided considerable insights into the behavior
345 of general neural networks, which is hard to analyze otherwise. Regarding (ii), in some easy tasks,
346 when early stopping is applied, existing algorithms for distributional shift can do better than ERM
347 [SKHL20]. However, as demonstrated in [GLP21, KSM$^+$21], in real applications these methods still
348 cannot significantly improve over ERM even with early stopping, so early stopping is not the ultimate
349 universal solution. Thus, though inevitably our results rely on some strong assumptions, we believe
350 that they provide important insights into the problems of existing methods and directions for future
351 work, which are significant contributions to the study of distributional shift problems.

## 7 Conclusion

353 In this work, we posit a broad class of what we call Generalized Reweighting (GRW) algorithms that
354 include popular approaches such as importance weighting, and Distributionally Robust Optimization
355 (DRO) variants, that were designed towards the task of learning models that are robust to distributional
356 shift. We show that when used to train overparameterized linear models or wide NN models, even this
357 very broad class of GRW algorithms does not improve over ERM, because they have the same implicit
358 biases. We also showed that regularization does not help if it is not large enough to significantly
359 lower the average training performance. Our results thus suggest to make progress towards learning
360 models that are robust to distributional shift, we have to either go beyond this broad class of GRW
361 algorithms, or design new losses specifically targeted to this class.

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
