# OpenReview forum: "Understanding Why Generalized Reweighting Does Not Improve Over ERM"
_NeurIPS.cc/2022/Conference — NeurIPS 2022 Submitted_

### Official Review · Reviewer_sLyL · 2022-07-11

**Rating:** 4
**Confidence:** 2
**Soundness:** 3 good
**Presentation:** 3 good
**Contribution:** 1 poor

**Summary:**

Focused on understanding the effect of reweighing strategy on improving the robustness of ERM to distributional shift, this paper demonstrates notably that iteratively updating the sample weights does not improve over ERM in an overparametrized setting. With similars results already derived for the method of fixed weights, the main contribution of this work is to confirm, in a more general setting, the fact that the solution obtained by gradient descent in the overparametrized regime converges to an interpolator independent of the samples weights.

**Questions:**

As the main contribution of this work concerns the generalization to the iterative reweighing setting, it would help to clarify the originality of this work to discuss the technical challenges brought by the iteratively updated sample weights in obtaining the theoretical results.

Furthermore, the interest of this study would be significantly increased by adding deeper results in the regularized case to shed light on the impact of the iteratively reweighed samples on the improvement of robustness.

**Limitations:**

The limitations of this work are well discussed. There does not seem to be any negative societal impact.

The contribution of this paper would be greatly enhanced if the authors could go further in exploring one of the directions mentioned in the article where the robustness is likely to improve.

**Strengths And Weaknesses:**

Strengths: Theoretical understanding of reweighing strategy under a more general framework that allows iterative updating of sample weights.

Weaknesses : As previous work already explained the ineffectiveness of sample reweighing in the overparametrized regime with the convergence of the gradient descent solution to an interpolator independent of the sample weights, the generalized results given in this paper do not seem to provide further insight into the effect of the reweighing approach.

---

> ### Author Response · Authors · 2022-08-01
> **Response to Reviewer sLyL**
>
> Thanks for your review! We would like to address your concerns as follows:
>
> We strongly disagree with the reviewer's comment that this work has no further insights into the effect of reweighting approaches, because we extend previous results to a great extent and in three ways: (i) Extend the results to neural networks; (ii) Extend the results to dynamic reweighting; (iii) Cover both regression and classification tasks. These extensions allow us to cover a much broader setting, including popular methods like DRO, Group DRO, etc.
>
> First, to our best knowledge, we are the first to prove these theoretical results on neural networks. All previous theoretical results comparing reweighting with ERM, including all those cited in the paper, are constrained to linear models. Moreover, proving the results for neural networks is much more difficult than linear models because it depends on the approximation theorem that approximates a NTK neural network with its linearized counterpart throughout the training process (Lemma 5).
>
> Second, to our best knowledge, all previous theoretical results comparing reweighting with ERM are constrained to approaches with fixed sample weights, and we are the first to extend this result to the broad GRW class where the sample weights are not fixed. GRW includes popular algorithms such as DRO and Group DRO which previous results didn't cover, and we believe that this is a very important improvement. Proving the results for dynamic GRW is much more difficult than fixed weight reweighting because the loss function is constantly changing. Most theoretical results in optimization are based on the assumption that the objective is fixed. When the objective is changing, proving the same results is very challenging.
>
> Third, we are the first to cover regression tasks. While [GLSS18] also studied regression tasks, they didn't study reweighting and distribution shift.
>
> We encourage the reviewer to read our proofs in Appendix D for our technical novelties. Provided that our work extends previous results in these three important ways with lots of technical novelties, we firmly believe that our work has sufficient originality and important technical contributions to this field, and we sincerely hope that the reviewer can rethink about the contributions of this work.
>
> Regarding the regularized case, we have already had two theorems about regularization (Theorem 6 and Theorem 9), which prove that regularized GRW cannot do better than ERM if the regularization is not big enough, which is also empirically observed in previous work [SKHL20].
>
> Regarding the future directions mentioned in the paper, we don't want to expand them in this work because we want to keep the main focus of this paper to explaining why existing GRW methods cannot do better than ERM.
>
> We hope that our response addresses your concerns. If so, it would be great if you can raise your rating.

---

> > ### Author Response · Authors · 2022-08-08
> > **Author Discussion**
> >
> > Dear reviewer,
> >
> > If you would like to have a discussion with us, we would be very happy to do so.
> >
> > Thanks,
> > Authors

---

### Official Review · Reviewer_8VCe · 2022-07-11

**Rating:** 7
**Confidence:** 3
**Soundness:** 3 good
**Presentation:** 3 good
**Contribution:** 3 good

**Summary:**

This paper shows that Generalized Reweighting (GRW), dynamically weighting each loss term of the empirical risk, used for training a model that is robust to distribution shift has almost no impact on the result of the optimization compared with the ordinary Empirical Risk Minimization (ERM) for regression and classification with linear models or wide neural networks. This implies that despite being a popular approach, GRW fails to improve over ERM in the cases that the authors consider.
The authors also prove that even with regularization that compromise the training loss by $\epsilon$, GRM only brings a change of $O(\sqrt{\epsilon})$ to the minimizer compared to ERM.
Their results suggest that learning robustly to distribution shift may need approaches other than GRM.


**Questions:**

- [HNSS18] also provides some theoretical, negative results similarly to this paper. Could the authors discuss the difference?
- Do the authors have any thoughts about how much difference we should expect in the result of optimization, $f^{(t)}$, from that of ERM in order to be distributionally robust? If we only need a small change from ERM to achieve robustness, the result for regularization in the paper may not be a negative one.


**Limitations:**

The paper clearly discusses limitations of the work about the theoretical assumptions.

**Strengths And Weaknesses:**

# Strengths
- The paper is well-written and provide intuitive explanations even for technical part.
- The paper provides theoretical evidence supporting previously reported empirical observations for failures in training distributionally robust models with the weighting approach.
- While existing theoretical work only handles linear models, the paper consider wide neural networks.
- As a technical contribution, the authors analysis corrects and extends that of Lee et al. (2019).
- Although the main results are negative ones for the GRW approach, the paper discusses implications about other possible approaches.

# Weaknesses
- The paper does not provide concrete algorithms that can avoid their negative results.
- The paper analyzes the asymptotic behavior of wide neural networks when the widths go to infinity, which may not reflect the behavior of neural networks commonly used in reality.
- The results with regularization are hard to interpret because the paper does not analyze how much change we need for truly robust training and we do not know the $O(\sqrt{\epsilon})$ impact is insufficient or not.

---

> ### Author Response · Authors · 2022-08-01
> **Response to Reviewer 8VCe**
>
> Thanks for your review! We would like to address your questions as follows:
>
> - "Does not provide concrete algorithms to avoid negative results": In fact, in Section 6.1, we have discussed three concrete ways of avoiding the negative results in this paper. However, we don't want to expand on them because we want to keep the focus of this paper to explaining why GRW cannot better than ERM. We would like to keep these algorithms to future work.
>
> - "Widths go to infinity": Yes, models used in reality are not infinitely wide. However, infinitely wide NTK neural networks have been widely used in theory papers to prove a lot of results that provide researchers and practitioners with valuable insights. As we discussed in Section 6.2, as a theory paper, our paper inevitably needs to rely on some strong assumptions, but we still believe that this work provides important insights into how to solve the problems of existing reweighting methods.
>
> - "How much regularization needed for robust models": This really depends on the specific tasks. In practice, different datasets and different model architectures need different levels of regularization, and there is no general result to cover all of them. The point of our results is to show that this regularization must be large enough, or it cannot do better than ERM.
>
> - "Difference from [HNSS18]": The main argument of this paper is that for classification tasks, the minimizer of the DRSL objective (also known as CVaR) is the same as ERM, because there is a monotonic relationship between the DRSL and the ERM objectives. This result is very different from the main argument of this paper, which is "GRW converges to the same interpolator (or max-margin classifier) as ERM" for both regression and classification tasks, because they have equivalent implicit biases, but not because of a monotonic relationship between the two objectives.
>
> - "How much difference of ERM could be sufficient": This also depends on the specific tasks. However, a lot of previous work such as [KSM+21] empirically showed that GRW cannot do better than ERM on a wide variety of realistic tasks. This is a well-known open problem in this field, and the goal of this work is to theoretically explain why this happens, and provide insights into possible ways of solving the problem.
>
> We hope that our response addresses your concerns.

---

> > ### Author Response · Authors · 2022-08-08
> > **Author Discussion**
> >
> > Dear reviewer,
> >
> > If you would like to have a discussion with us, we would be very happy to do so.
> >
> > Thanks,
> > Authors

---

### Official Review · Reviewer_XtEr · 2022-07-12

**Rating:** 5
**Confidence:** 3
**Soundness:** 2 fair
**Presentation:** 3 good
**Contribution:** 2 fair

**Summary:**

As a theoretical work, this paper aims to understand why generalized reweighting does not improve over ERM in the setting of distribution shift. Specifically, it considers the overparameterized linear model and wide neural networks under gradient descent, and prove the same solution between generalized reweighting and ERM methods via the implicit bias of the gradient descent algorithm.  Then, it claims that these two method families should share the same performance empirically. Technically, it mainly utilizes the neural tangent kernel theory for the wide neural networks.

**Questions:**

1. The authors mentioned that the reweighting and ERM algorithms converge to the unique interpolator. But which one? Does the interpolator have some special property?
2. Although this paper mainly considers the distribution shift problem, I haven't seen its key difference or point with the classical setting. Please give more explanations why ERM is not suitable for this case.
3. When theoretically comparing with empirical performance of two or many learning algorithms, generalization analysis is preferred. While this paper focuses on the optimization properties via implicit bias of gradient descent, implicit bias can also offer insight for generalization. Thus, what is the generalization performance of the learning algorithms considered in this paper?

**Limitations:**

The authors have mentioned some limitations in this paper. But, I still have the following suggestions.
1. The generalization analysis can provide more insights into the empirical performance.
2. More experiments in real settings can support the theoretical results.

**Strengths And Weaknesses:**

Strengths:
1. This work is motivated by the empirical results that the reweighting methods often have no superiority over ERM in the distribution shift problem. In my view, its theoretical investigation is valuable.
2. The claim is partially supported by formal theoretical results.

Weaknesses:
1. The considered setting is rather limited as previous work has considered the linear model case. Besides, although this paper considers the wide neural networks via the connection between the kernel method and neural networks (i.e., neural tangent kernel theory), it still has a large gap in practice.
2. The results are not surprising and interesting as mentioned in 1. The key reason why generalized reweighting and ERM converge to the same solution is the implicit bias of the optimization algorithms (i.e., gradient descent in this paper) for overparameterized models. However, this paper does not highlight the key effects of the overparameterized model case. Thus, I am concerned that the authors may overclaim the results.
3. In practice, regularization and early stopping are usually used to improve the generalization performance. However, this paper neglects them.

---

> ### Author Response · Authors · 2022-08-01
> **Response to Reviewer XtEr**
>
> Thanks for your review! We would like to address your questions and concerns as follows:
>
> Concerns in the "Weaknesses" part:
>
> 1. We cannot agree with the reviewer's comment that "the considered setting is rather limited". We extend previous results to a great extent and in three ways: (i) Extending the results to NTK neural networks, while all previous theoretical results comparing reweighting to ERM are constrained to linear models; (ii) Extending the results to dynamic GRW where the model weights can change, while all such previous theoretical results are constrained to fixed weight reweighting; (iii) Covering regression tasks for the first time. These extension are very complicated, and we encourage the reviewer to read our proofs in Appendix D for our technical novelties. We consider the very broad class of GRW algorithms for linear models and NTK neural networks and for both regression and classification tasks, which we believe is a very broad and general setting and is hard to be called "limited".
>
> 2. The models we use to prove our results, including linear models and NTK neural networks, are the most widely used general models in existing theory papers. While there is a gap between general models and practical models, the insights obtained from the results for these general models have been proved to be very useful for practical models too, which is the value of theory papers.
>
> 3. We cannot agree with the reviewer's comment that "these results are not surprising". When previous papers [BL19, SKHL20, KSM+21] empirically observed that GRW couldn't improve over ERM, they found this phenomenon very surprising and baffling, and our paper explains the reason behind this baffling phenomenon, which has long been an open problem in this field. Regarding the effect of overparameterization, we have discussed this point in Section 6.2.
>
> 4. We strongly disagree with the reviewer's comment that we "neglect regularization and early stopping". In fact, we have two theorems about the effect of regularization (Theorem 6 and Theorem 9), and in Section 6.2 we discuss the problems of early stopping.
>
> Questions:
>
> 1. "What is the unique interpolator?" For regression, the unique interpolator is the interpolator that is the closest to the initial point (see [GLSS18]). We will briefly talk about its properties in the updated version of this paper. For classification, it is the max-margin classifier (Eqn. (17)).
>
> 2. "ERM cannot lead to distributionally robust models", or "ERM leads to unfair models and cannot learn minority classes well", is a well-known problem in this field. That is why a rich body of work studies how to train robust models, and proposes methods like importance weighting, DRO, Group DRO, and many others. However, a number of previous papers empirically observed that these "robust training" methods do not really improve over ERM on a variety of realistic tasks, and our paper theoretically explains why this is the case. Moreover, in Section 6.1 we point out 3 possible ways of fixing the problems of existing methods.
>
> 3. In this paper we proved that GRW has an equivalent implicit bias to ERM, which means that GRW and ERM should have the same OOD generalization performance. If ERM couldn't generalize well on a certain task, then neither could GRW. Meanwhile, as we said in the previous point, "ERM leads to unfair models and cannot learn minority classes well" is a well-known problem, which implies that GRW will also lead to unfair models and cannot learn minority classes well, i.e. poor OOD generalization performances.
>
> 4. Experiments in real settings have already been done in a number of previous papers [BL19, SKHL20, GLP21, KSM+21] as we mentioned in lines 36-43 and there is no need to do them again. The focus of this paper is to theoretically explain the empirical observations in these previous work.
>
>
> We hope that our response addresses your questions and concerns. If so, it would be great if you can raise your rating.

---

> > ### Author Response · Authors · 2022-08-08
> > **Author Discussion**
> >
> > Dear reviewer,
> >
> > If you would like to have a discussion with us, we would be very happy to do so.
> >
> > Thanks,
> > Authors

---

### Official Review · Reviewer_85jH · 2022-07-12

**Rating:** 4
**Confidence:** 4
**Soundness:** 3 good
**Presentation:** 3 good
**Contribution:** 3 good

**Summary:**

This manuscript demonstrates that the popular approaches, such as importance weighting, and Distributionally Robust Optimization (DRO) variants, cannot really improve the distributional drift problem over Empirical risk minimization (ERM). To achieve this goal, this manuscript presents a Generalized Reweighting (GRW) (the above popular algorithms are its special cases). When training overparameterized linear models or wide NN models, the result for GRW is very close to that for ERM. Moreover, it also demonstrates a small regularization that needs to not greatly affect the empirical training accuracy does not help.

**Questions:**

The theoretical analysis is built on linear models. To extend to neural networks, the manuscripts utilizes existing the neural tangent kernel (NTK) theory to take networks as an approximately linear model. It is known that there is a gap between NTK networks and the practical networks. The width of NTK networks should be infinite; the NTK networks are commonly simple fully-connected networks;  he optimization algorithms should be simple stochastic gradient descent (SGD). All these conditions cannot be satisfied in real-world tasks. Moreover, Assumption assumes the first-order of the active function is Lipschitz continuous. We know it is common to apply ReLU as the active function in the modern networks, which is not satisfied this assumption. In summary, these ideal assumptions will influence the validity of the theoretical analysis.

**Limitations:**

There is no potential negative societal impact for this manuscript.

**Strengths And Weaknesses:**

Observing that some works uncovered the popular approaches that aim to address distributional drift do not significantly improve over ERM, the authors try to theoretically prove this empirical phenomenon. Moreover, according to the theoretical conclusion, the authors also provide some potential approaches that can alleviate the distributional drift problem.  and it might fundamentally change this direction for distributionally robust approaches.

---

> ### Author Response · Authors · 2022-08-01
> **Response to Reviewer 85jH**
>
> Thanks for your review! We would like to address your concerns as follows:
>
> 1. "There is a gap between NTK networks and practical networks": The models we use to prove our results, including linear models and NTK neural networks, are the most widely used general models in existing theory papers. While there is a gap between general models and practical models, the insights obtained from the results for these general models have been proved to be very useful for practical models too, which is the value of theory papers. In this paper, we theoretically explain a baffling phenomenon observed in previous work, pinpoint a problem that exists in existing GRW methods, and finally, like the reviewer said, aim to "change this direction for distributionally robust approaches", which we believe is a great contribution to this field. As we have said in Section 6.2, as a theory paper, our work inevitably needs to depend on some assumptions, but it by no means implies that our results cannot provide useful insights for practical situations.
>
> 2. Moreover, our work extends the results in previous work in many ways. (i) To our best knowledge all previous theoretical results that compare reweighting with ERM (including all the citations in this paper) are constrained to linear models, and we are the first to extend this line of analysis to neural networks, which is much more technically complicated. (ii) all such previous results are constrained to fixed weight reweighting, and we are the first to extend these results to dynamic GRW which covers most popular methods like DRO. (iii) we are the first to cover both regression and classification tasks. We believe that these are important technical contributions to this field.
>
> 3. "The optimization is always SGD": In fact, our results can be easily extended to other first-order optimization methods. For example, Theorem 1 is true for any gradient method that converges, including AdaGrad, Adam and many more. We thank the reviewer for bringing this up, and will briefly talk about this in the updated version, but please understand that we cannot address each optimization method one by one.
>
> We hope that our response addresses your concerns. If so, it would be great if you can raise your rating.

---

> > ### Author Response · Authors · 2022-08-08
> > **Author Discussion**
> >
> > Dear reviewer,
> >
> > If you would like to have a discussion with us, we would be very happy to do so.
> >
> > Thanks,
> > Authors

---

### Author Response · Authors · 2022-08-09
**Author Discussion**

Dear Reviewers,

If you would like to have a last minute discussion with us, we would be very happy to discuss.

Thanks,
Authors

---

### Meta-Review · Area_Chair_zUSg · 2022-08-27

**Recommendation:** Reject
**Confidence:** Less certain

**Metareview:**

The paper investigates generalized reweighting schemes that assign to each sample a weight during each iteration of GD over the empirical error. In the case of over-parameterized linear models and linearly independent samples it is shown that the final(= GD with infinite trajectory) ERM solution is not affected by the reweighting as long as the weights converge. These results are extended to the NTK regime with 0-initialization of the last layer under the assumption that the samples in the initial NTK-feature space are linearly independent.

Compared to previous papers it allows different weights at each iteration and applies the setup to neural networks. However, as the weights need to converge (see Assumption 1) and only the NTK-regime (which is rather close to the previously considered linear case) is treated, the novelty is somewhat limited. In addition, it would have been nice to see interesting examples, in which the assumptions are actually satisfied.

Although I tend to vote for rejection, I think this paper should to be compared to other papers of similar strength in order to make a final decision.

**Award:**

No

---

### Decision · Program_Chairs · 2022-09-14

Reject